# TENSOR-BASED SKETCHING METHOD FOR THE LOW-RANK APPROXIMATION OF DATA STREAMS

**Cuiyu Liu**[1], **Chuanfu Xiao**[2 3], **Mingshuo Ding**[1], **Chao Yang**[2 3*]

[1]Academy for Advanced Interdisciplinary Studies, Peking University, Beijing, China
[2]School of Mathematical Sciences, Peking University, Beijing, China
[3]Changsha Institute for Computing and Digital Economy, Changsha, China
`2101213203@stu.pku.edu.cn`,
`{chuanfuxiao,dingmingshuo,chao_yang}@pku.edu.cn`

## ABSTRACT

Low-rank approximation in data streams is a fundamental and significant task in computing science, machine learning and statistics. Multiple streaming algorithms have emerged over years and most of them are inspired by randomized algorithms, more specifically, sketching methods. However, many algorithms are not able to leverage information of data streams and consequently suffer from low accuracy. Existing data-driven methods improve accuracy but the training cost is expensive in practice. In this paper, from a subspace perspective, we propose a tensor-based sketching method for low-rank approximation of data streams. The proposed algorithm fully exploits the structure of data streams and obtains quasi-optimal sketching matrices by performing tensor decomposition on training data. A series of experiments are carried out and show that the proposed tensor-based method can be more accurate and much faster than the previous work.

## 1 INTRODUCTION

There are many scenarios that require batch or real-time processing of data streams arising from, e.g., video (Cyganek & Woźniak, 2017; Das, 2021), signal flow (Cichocki et al., 2015; Sidiropoulos et al., 2017), hyperspectral images (Wang et al., 2017; Zhang et al., 2019) and numerical simulations (Zhang et al., 2022; Larcher & Klein, 2019). A data stream can be seen as an ordered sequence of data continuously generated from one or several distributions (Muthukrishnan, 2005; Indyk et al., 2019), and the data per time slot can be usually represented as a matrix. Therefore, most of the processing methods of data streams can be considered as operations on matrices, such as matrix multiplications, linear system solutions and low-rank approximation. Wherein, low-rank matrix approximation plays an important role in practical applications, such as independent component analysis (ICA) (Stone, 2002; Hyvärinen, 2013), principle component analysis (PCA) (Karamizadeh et al., 2020; Jolliffe & Cadima, 2016), image denoising (Guo et al., 2015; Zhang et al., 2019).

In this work, we consider low-rank approximation of matrices from a data stream. Specifically, let $\{\boldsymbol{A}_d \in \mathbb{R}^{m \times n}\}_{d=1}^{D}$ be matrices from a data stream $\mathcal{D}$, then the low-rank approximation in $\mathcal{D}$ can be described as:

$$\min_{\boldsymbol{B}_d} \|\boldsymbol{A}_d - \boldsymbol{B}_d\|_F, \ \ \text{s.t. } \text{rank}(\boldsymbol{B}_d) \leq r, \tag{1.1}$$

where $d = 1, 2, \cdots, D$, $\| \cdot \|_F$ represents the Frobenius norm, and $r \in \mathbb{Z}_+$ is a user-specified target rank.

**Related work.** A direct approach to solve problem 1.1 is to calculate the truncated rank-$r$ singular value decomposition (SVD) of $\boldsymbol{A}_d$ in turn, and the Eckart-Young theorem ensures that it is the best low-rank approximation (Eckart & Young, 1936). However, it is too expensive to one by one calculate the truncated rank-$r$ SVD of $\boldsymbol{A}_d$ for all $d = 1, 2, \cdots, D$, particularly when $m$ or $n$ is large. To address this issue, many sketching algorithms have emerged such as the SCW algorithm (Sarlos, 2006; Clarkson & Woodruff, 2009; 2017). Unfortunately, a notable weakness of sketching

---

*Correspondence to Chao Yang.

algorithms is that they achieve higher error than the best low-rank approximation, especially when the sketching matrix is generated randomly from some distribution, such as Gaussian, Cauchy, or Rademacher distribution (Indyk, 2006; Woolfe et al., 2008; Clarkson & Woodruff, 2009; Halko et al., 2011; Clarkson & Woodruff, 2017). To improve accuracy, a natural idea is to perform a preprocessing on the past data (seen as a training set) in order to better handle the future input matrices (seen as a test set). This approach, which is often called the data-driven approach, has gained more attention lately. For low-rank approximation, the pioneer of this work was (Indyk et al., 2019), who proposed a learning-based method, that we henceforth refer to as IVY. In the IVY method, the sketching matrix is set to be sparse, and the values of non-zero entries are learned instead of setting them randomly as classical methods do. Specifically, learning is done by stochastic gradient descent (SGD), by optimizing a loss function that portrays the quality of the low-rank approximation obtained by the SCW algorithm as mentioned above. To improve accuracy, (Liu et al., 2020) followed the line of IVY by additionally optimizing the location of the non-zero entries of the sketching matrix $S$, not only their values. Recently, (Indyk et al., 2021) proposed a Few-Shot data-driven low-rank approximation algorithm, and their motivation is to reduce the training time cost of (Indyk et al., 2019). Wherein, they proposed an algorithm namely FewShotSGD by minimizing a new loss function that measures the distance in subspace between the sketching matrix $S$ and all left-SVD factor matrices of the training matrices, with SGD. However, these data-driven approaches all involve learning mechanisms, which require iterations during the optimization process. This raises a question: can we design an efficient method, such as a non-iterative method, to get a better sketching matrix with both short training time and high approximation quality? It would be an important step for the development of data-driven methods, especially in scenarios requiring low latency.

**Our contributions.** In this work, we propose a new data-driven approach for low-rank approximation of data streams, motivated by a subspace perspective. Specifically, we observe that a perfect sketching matrix $S \in \mathbb{R}^{k \times m}$ should be close to the top-$k$ subspace of $U^d$, where $U^d$ is the left-SVD factor matrix of $A_d$. Due to the relevance of matrices in a data stream, it allows us to develop a new sketching matrix $S$ to approximate the top-$k$ subspace of $U^d$ for all $d = 1, \cdots, D$. Perhaps the heavy learning mechanisms can be eliminated. In fact, our approach attains the sketching matrix by minimizing a new loss function which is a relaxation of that in IVY. The most important thing is that we can get the minimization of this loss function by tensor decomposition on the training set, which is non-iterative. We refer to this method as tensor-based method. As an extension of the main approach, we also develop the two-sided tensor-based algorithm, which involves two sketching matrices $S$, $W$. These two sketching matrices can be obtained simultaneously by performing tensor decomposition once. Both algorithms are significantly faster and more accurate than the previous data-driven approaches.

## 2 PRELIMINARIES

**The SCW algorithm.** Randomized SVD is an efficient algorithm for computing the low-rank approximation of matrices from a data stream. For example, the SCW algorithm, proposed by Sarlos, Clarkson and Woodruff (Sarlos, 2006; Clarkson & Woodruff, 2009; 2017), is a classical randomized SVD algorithm. The algorithm only computes the SVD of the compressed matrices $SA$ and $AV$, and its time cost is $\mathcal{O}(r^2(m+n))$ when we set $k = \mathcal{O}(r)$. The detailed procedure is shown in Algorithm 1.

---

**Algorithm 1** The SCW algorithm (Sarlos, 2006; Clarkson & Woodruff, 2009; 2017).

---

**Input:** Matrix $A \in \mathbb{R}^{m \times n}$, sketching matrix $S \in \mathbb{R}^{k \times m}$, and target rank $r < \min\{m, n\}$
 1: $\sim, \sim, V^T \leftarrow$ full SVD of $SA$
 2: $[AV]_r \leftarrow$ truncated rank-$r$ SVD of $AV$
 3: $\hat{A} \leftarrow [AV]_r V^T$
**Output:** Low-rank approximation of $A$: $\hat{A}$

---

In (Clarkson & Woodruff, 2009), it is proved that if $S$ satisfies the property of Johnson-Lindenstrauss Lemma, $k = O(r \log(1/\delta)/\varepsilon)$ suffices the output $\hat{A}$ to satisfy $\|A - \hat{A}\|_F \le (1 + \varepsilon)\|A - [A]_r\|_F$

with probability $1-\delta$. Therefore, the approximation quality of the SCW algorithm is highly dependent on the choice of the sketching matrix $\boldsymbol{S}$. In general, the randomly generated sketching matrix does not meet the accuracy requirements when we handle problems in a data stream, so can we design a new $\boldsymbol{S}$ by utilizing the information of the data stream? This is the motivation of data-driven approaches.

**The IVY algorithm.** In (Indyk et al., 2019), the sketching matrix $\boldsymbol{S}$ is initialized by a sparse random sign matrix as described in (Clarkson & Woodruff, 2009). The location of the non-zero entries is fixed, while the values are optimized with SGD via the loss function as follow.

$$\min_{\boldsymbol{S} \in \mathbb{R}^{k \times m}} \sum_{\boldsymbol{A} \in \mathcal{D}_{\text{train}}} \|\boldsymbol{A} - \text{SCW}(\boldsymbol{A}, \boldsymbol{S}, r)\|_F^2, \tag{2.1}$$

where $\mathcal{D}_{\text{train}}$ is the training set sampled from the data stream $\mathcal{D}$. This requires computing the gradient of the SCW operator, which involves the SVD implementation (line 1 and 2 in Algorithm 1). IVY uses a differential but inexact SVD based on the power method, and (Liu et al., 2020) suggested that the SVD in PyTorch is also feasible and much more efficient.

**The Few-Shot algorithm.** In (Indyk et al., 2021), $\boldsymbol{S}$ is initialized the same way as IVY, and the location of non-zero entries remains, too. The difference is that the authors optimize the non-zero values by letting $\boldsymbol{S}$ to approximate the left top-$r$ subspace of a few training matrices. Wherein, the proposed algorithm namely FewShotSGD minimizes the following loss function:

$$\min_{\boldsymbol{S} \in \mathbb{R}^{k \times m}} \sum_{\boldsymbol{U} \in \mathcal{U}_{\text{train}}} \|\boldsymbol{U}_r^T \boldsymbol{S}^T \boldsymbol{S} \boldsymbol{U} - \boldsymbol{I}_0\|_F^2, \tag{2.2}$$

where $\mathcal{U}_{\text{train}} = \{\boldsymbol{U} : \boldsymbol{A} = \boldsymbol{U} \boldsymbol{\Sigma} \boldsymbol{V}^T \text{ of all } \boldsymbol{A} \in \mathcal{D}_{\text{train}}\}$, $\boldsymbol{U}_r$ denotes a matrix containing the first $r$ columns of $\boldsymbol{U}$, and $\boldsymbol{I}_0 \in \mathbb{R}^{r \times n}$ has zero entries except that $(\boldsymbol{I}_0)_{i,i} = 1$ for $i = 1, \cdots, r$.

As shown in (Indyk et al., 2021), the goal of FewShotSGD is to get the sketch which preserves the left top-$r$ subspace of all matrices $\boldsymbol{A} \in \mathcal{D}_{\text{train}}$ well and meanwhile is orthogonal to their bottom-$(n-r)$ subspace. This raises a question: can we directly obtain a subspace that is close to the top-$r$ subspace of all matrices $\boldsymbol{A} \in \mathcal{D}_{\text{train}}$ are required to be viewed as a whole, i.e., a third-order tensor. For illustration, we introduce some basics about tensor before presenting our method.

**Tensor basics.** For convenience, we only consider the third-order tensor $\boldsymbol{\mathcal{A}} \in \mathbb{R}^{m \times n \times D}$, and $\boldsymbol{\mathcal{A}}_{i,j,d}$ represents the $(i, j, d)$-th entry of $\boldsymbol{\mathcal{A}}$. The Frobenius norm of $\boldsymbol{\mathcal{A}}$ is defined as $\|\boldsymbol{\mathcal{A}}\|_F = \sqrt{\sum_{i,j,d} \boldsymbol{\mathcal{A}}_{i,j,d}^2}$. The mode-$n$ ($n = 1, 2, 3$) matricization of $\boldsymbol{\mathcal{A}}$ is to reshape it to a matrix $\boldsymbol{A}_{(n)}$. For example, the mode-1 matricization of $\boldsymbol{\mathcal{A}}$ is $\boldsymbol{A}_{(1)} \in \mathbb{R}^{m \times nD}$ satisfying $(\boldsymbol{A}_{(1)})_{i,1+(j-1)n+(d-1)mn} = \boldsymbol{\mathcal{A}}_{i,j,d}$. The 1-mode product of $\boldsymbol{\mathcal{A}}$ and a matrix $\boldsymbol{S} \in \mathbb{R}^{k \times m}$ is denoted as $\boldsymbol{\mathcal{B}} = \boldsymbol{\mathcal{A}} \times_1 \boldsymbol{S} \in \mathbb{R}^{k \times n \times D}$, which satisfies $\boldsymbol{\mathcal{B}}_{s,j,d} = \sum_{i=1}^m \boldsymbol{\mathcal{A}}_{i,j,d} \boldsymbol{S}_{s,i}$. Tucker decomposition (Tucker, 1966) is one format of tensor decomposition, which is also called higher-order singular value decomposition (HOSVD) (Lathauwer et al., 2000). It decomposes a tensor into a set of factor matrices and one small core tensor of the same order. For $\boldsymbol{\mathcal{A}} \in \mathbb{R}^{m \times n \times D}$, its Tucker decomposition is

$$\boldsymbol{\mathcal{A}} = \boldsymbol{\mathcal{G}} \times_1 \boldsymbol{U} \times_2 \boldsymbol{V} \times_3 \boldsymbol{W},$$

where $\boldsymbol{U} \in \mathbb{R}^{m \times r_1}, \boldsymbol{V} \in \mathbb{R}^{n \times r_2}, \boldsymbol{W} \in \mathbb{R}^{D \times r_3}$ are the column orthogonal factor matrices, $\boldsymbol{\mathcal{G}} \in \mathbb{R}^{r_1 \times r_2 \times r_3}$ is the core tensor, and $(r_1, r_2, r_3)$ is called the multilinear-rank of $\boldsymbol{\mathcal{A}}$. There are two important variations of Tucker decomposition, i.e., Tucker1 and Tucker2 (Kolda & Bader, 2009) (1 or 2 modes of $\boldsymbol{\mathcal{A}}$ are decomposed), which can be represented as $\boldsymbol{\mathcal{A}} = \boldsymbol{\mathcal{G}} \times_1 \boldsymbol{U}$ and $\boldsymbol{\mathcal{A}} = \boldsymbol{\mathcal{G}} \times_1 \boldsymbol{U} \times_2 \boldsymbol{V}$, respectively.

## 3 TENSOR-BASED SKETCHING METHOD

In this section, we present our idea and method for low-rank approximation in data streams. The goal is to employ the given training set to get the sketch $\boldsymbol{S}$, inspired by IVY (Indyk et al., 2019) and FewShotSGD (Indyk et al., 2021).

### 3.1 TENSOR-BASED ALGORITHM

Our main algorithm, the tensor-based algorithm, is also a data-driven algorithm for low-rank approximation in data streams. Instead of minimizing the loss 2.1 in IVY, we consider a different loss, motivated by a subspace perspective. This loss function is easier to optimize than 2.1 since to get its minimization, only a Tucker1 decomposition is required, without learning mechanisms.

Let $\mathcal{D}_{\text{train}} = \{\boldsymbol{A}_{d'} \in \mathbb{R}^{m \times n}\}_{d'=1}^{D'}$, and the loss function we consider is

$$\min_{\boldsymbol{S} \in \mathbb{R}^{k \times m}} \sum_{\boldsymbol{A}_{d'} \in \mathcal{D}_{\text{train}}} \|\boldsymbol{A}_{d'} - \boldsymbol{S}^T \boldsymbol{S} \boldsymbol{A}_{d'}\|_F^2, \ \text{ s.t. } \boldsymbol{S} \boldsymbol{S}^T = \boldsymbol{I}_k. \tag{3.1}$$

Using the row-wise orthogonality of $\boldsymbol{S}$, we have $\|\boldsymbol{A}_{d'} - \boldsymbol{S}^T \boldsymbol{S} \boldsymbol{A}_{d'}\|_F^2 = \|\boldsymbol{A}_{d'}\|_F^2 - \|\boldsymbol{S} \boldsymbol{A}_{d'}\|_F^2$. Let $\mathcal{A} \in \mathbb{R}^{m \times n \times D'}$ be a third-order tensor satisfying $\mathcal{A}_{:,:,d'} = \boldsymbol{A}_{d'}$. To minimize 3.1, it is equivalent to solve

$$\max_{\boldsymbol{S} \in \mathbb{R}^{k \times m}} \sum_{\boldsymbol{A}_{d'} \in \mathcal{D}_{\text{train}}} \|\boldsymbol{S} \boldsymbol{A}_{d'}\|_F^2 \iff \max_{\boldsymbol{S} \in \mathbb{R}^{k \times m}} \|\boldsymbol{S} \boldsymbol{A}_{(1)}\|_F^2, \ \text{ s.t. } \boldsymbol{S} \boldsymbol{S}^T = \boldsymbol{I}_k, \tag{3.2}$$

where $\boldsymbol{A}_{(1)} = [\boldsymbol{A}_1 | \boldsymbol{A}_2 | \cdots | \boldsymbol{A}_{D'}]$ is the mode-1 matricization of $\mathcal{A}$. Further, as shown in (Kolda & Bader, 2009), problem 3.2 is equivalent to

$$\min_{\boldsymbol{S} \in \mathbb{R}^{k \times m}} \|\mathcal{A} - \mathcal{G} \times_1 \boldsymbol{S}^T\|_F^2$$
$$\text{s.t. } \mathcal{G} \in \mathbb{R}^{k \times n \times D'}, \ \boldsymbol{S} \boldsymbol{S}^T = \boldsymbol{I}_k. \tag{3.3}$$

This is a Tucker1 decomposition of $\mathcal{A}$ along mode-1. Let $\boldsymbol{A}_{(1)} = \boldsymbol{U}^{(1)} \boldsymbol{\Sigma}^{(1)} (\boldsymbol{V}^{(1)})^T$ be the SVD of $\boldsymbol{A}_{(1)}$. The optimal sketch $\boldsymbol{S}^*$ for problem 3.3 is $(\boldsymbol{U}^{(1)})_k^T$, where $(\boldsymbol{U}^{(1)})_k$ is a matrix composed of the first $k$ columns in $\boldsymbol{U}^{(1)}$ (refer to (Kolda & Bader, 2009)). We use the optimal $\boldsymbol{S}^*$ as input of SCW, and get the output of SCW as the low-rank approximation. The tensor-based algorithm is summarized in Algorithm 2.

The motivation behind this choice of loss function is the theorem below, which illustrates the relationship between our loss function 3.1 and that in IVY.

**Theorem 1.** *Let $\boldsymbol{A}_{d'} \in \mathbb{R}^{m \times n}$ be a matrix from the training set, and $\mathcal{A} \in \mathbb{R}^{m \times n \times D'}$ be a third-order tensor satisfying $\mathcal{A}_{:,:,d'} = \boldsymbol{A}_{d'}$. Given the target rank $r \in \mathbb{Z}_+$, and a row-wise orthogonal matrix $\boldsymbol{S} \in \mathbb{R}^{k \times m}$, for any positive integer $k > r$, we have*

$$\sum_{d'=1}^{D'} \|\boldsymbol{A}_{d'} - \text{SCW}(\boldsymbol{A}_{d'}, \boldsymbol{S}, r)\|_F^2 \le \|\mathcal{A}\|_F^2 - \|[\boldsymbol{S} \boldsymbol{A}_{(1)}]_r\|_F^2, \tag{3.4}$$

*where $\boldsymbol{A}_{(1)} = [\boldsymbol{A}_1 | \boldsymbol{A}_2 | \cdots | \boldsymbol{A}_{D'}]$ is the mode-1 matricization of $\mathcal{A}$. Furthermore, with this relaxation, problem 2.1 can be converted to our proposed problem 3.3.*

Theorem 1 justifies the rationality of our choice of the loss function. Below we give an analysis that using the sketch obtained by problem 3.3, the SCW computes a good low-rank approximation of $\boldsymbol{A}$.

**Analysis.** In fact, our idea is similar to that in (Indyk et al., 2021) — both choosing the sketch $\boldsymbol{S}$ to approximate the top-$r$ row subspace of matrices in $\mathcal{D}_{\text{train}}$. Let $\boldsymbol{U} \boldsymbol{\Sigma} \boldsymbol{V}^T$ be the SVD of $\boldsymbol{A}$, where $\boldsymbol{A}$ is a matrix in $\mathcal{D}_{\text{train}}$. Since there is strong relevance among matrices in $\mathcal{D}_{\text{train}}$, it makes sense to assume that $\boldsymbol{S}$ obtained by 3.3 is close in space to $\boldsymbol{U}_k$ ($k > r$), where $\boldsymbol{U}_k$ is a matrix composed of the first $k$ columns of $\boldsymbol{U}$. In a special case where all matrices in $\mathcal{D}_{\text{train}}$ are the same, i.e., $\boldsymbol{A}_{d'} = \boldsymbol{A}$ for $d' = 1, \cdots, D'$, using $\boldsymbol{S}$ obtained by 3.3, we have $\|\boldsymbol{U}_k \boldsymbol{U}_k^T - \boldsymbol{S}^T \boldsymbol{S}\|_F^2 = 0$. Theorem 2 shows that using $\boldsymbol{S}$ computed by tensor-based algorithm, the SCW gives a good low-rank approximation of $\boldsymbol{A}$ in a data stream.

**Theorem 2.** *Let $\boldsymbol{U} \boldsymbol{\Sigma} \boldsymbol{V}^T$ be the SVD of $\boldsymbol{A} \in \mathbb{R}^{m \times n}$, and $\boldsymbol{U}_k$ be a matrix composed of the first $k$ columns of $\boldsymbol{U}$. Given a row-wise orthogonal sketching matrix $\boldsymbol{S} \in \mathbb{R}^{k \times m}$ satisfying $\|\boldsymbol{U}_k \boldsymbol{U}_k^T - \boldsymbol{S}^T \boldsymbol{S}\|_F^2 < \varepsilon$, then we have*

$$\|\boldsymbol{A} - \text{SCW}(\boldsymbol{A}, \boldsymbol{S}, r)\|_F^2 - \|\boldsymbol{A} - [\boldsymbol{A}]_r\|_F^2 < \mathcal{O}(\varepsilon) \|\boldsymbol{A}\|_F^2. \tag{3.5}$$

The proofs of Theorem 1 and 2 are provided fully in the Appendix.

---

**Algorithm 2** The tensor-based algorithm for low-rank approximation of the data stream $\mathcal{D}$.

---

**Input:** Test matrix $\boldsymbol{A}$, training set $\{\boldsymbol{A}_{d'} \in \mathbb{R}^{m \times n}\}_{d'=1}^{D'}$, rank $r \leq \min\{m, n\}$, # rows of the sketching matrix $k$.
1: Tensorization: $\mathcal{A} \in \mathbb{R}^{m \times n \times D'} \leftarrow \{\boldsymbol{A}_{d'}\}_{d'=1}^{D'}$
2: $\boldsymbol{S} \leftarrow$ Tucker1 decomposition of $\mathcal{A}$ along the mode-1
3: $\hat{\boldsymbol{A}} \leftarrow \text{SCW}(\boldsymbol{A}, \boldsymbol{S}, r)$
**Output:** Low-rank approximation of $\boldsymbol{A}$: $\hat{\boldsymbol{A}}$

---

## 3.2 Two-sided tensor-based algorithm

The two-sided tensor-based algorithm is an extension of the tensor-based algorithm in Section 3.1. The motivation is that if we compute the Tucker2 decomposition of $\mathcal{A}$ mentioned in Theorem 1, two sketching matrices $\boldsymbol{S}$ and $\boldsymbol{W}$, would be computed at once. This means that besides using $\boldsymbol{S}$ for row space compression, we can use $\boldsymbol{W}$ to compress the column space of $\boldsymbol{A}$, too. To be clear, we consider

$$\min_{\boldsymbol{S} \in \mathbb{R}^{k \times m}, \boldsymbol{W} \in \mathbb{R}^{l \times n}} \|\mathcal{A} - \mathcal{G} \times_1 \boldsymbol{S}^T \times_2 \boldsymbol{W}^T\|_F^2$$
$$\text{s.t. } \mathcal{G} \in \mathbb{R}^{k \times l \times D'}, \quad (3.6)$$
$$\boldsymbol{S}\boldsymbol{S}^T = \boldsymbol{I}_k, \ \boldsymbol{W}\boldsymbol{W}^T = \boldsymbol{I}_l.$$

Unlike 3.3, the exact solution of problem 3.6 has no explicit form, but can be efficiently approximated by an alternating iteration algorithm, namely higher-order orthogonal iteration (HOOI) (Lathauwer et al., 2000; Kolda & Bader, 2009). We present the HOOI algorithm in the Appendix. However, the SCW algorithm requires only one sketching matrix for computing low-rank approximation. As a result, a new sketching algorithm for two sketches is required.

**Two-sided SCW.** To this end, we develop a new algorithm for low-rank approximation based on the SCW algorithm, which we call two-sided SCW. It is worth mentioning that the full SVD in line 1 of Algorithm 1 is used for orthogonalization, thus it can be replaced with QR decomposition to improve the computational efficiency. With this in mind, the procedure of the two-sided SCW that we design is as shown in Algorithm 3.

---

**Algorithm 3** The two-sided SCW algorithm.

---

**Input:** Matrix $\boldsymbol{A} \in \mathbb{R}^{m \times n}$, sketching matrices $\boldsymbol{S} \in \mathbb{R}^{k \times m}$ and $\boldsymbol{W} \in \mathbb{R}^{l \times n}$, and target rank $r < \min\{m, n\}$
1: $\boldsymbol{Q}, \sim \leftarrow$ QR decomposition of $\boldsymbol{A}^T \boldsymbol{S}^T$
2: $\boldsymbol{P}, \sim \leftarrow$ QR decomposition of $\boldsymbol{A}\boldsymbol{W}^T$
3: $[\boldsymbol{P}^T \boldsymbol{A} \boldsymbol{Q}]_r \leftarrow$ truncated rank-$r$ SVD of $\boldsymbol{P}^T \boldsymbol{A} \boldsymbol{Q}$
4: $\boldsymbol{P}[\boldsymbol{P}^T \boldsymbol{A} \boldsymbol{Q}]_r \boldsymbol{Q}^T \leftarrow$ low-rank approximation of $\boldsymbol{A}$
**Output:** Low-rank approximation of $\boldsymbol{A}$: $\hat{\boldsymbol{A}}$

---

Clearly, Algorithm 3 is more efficient than the original SCW when $m$, $n$ are both large. The truncated SVD only needs to be done on $\boldsymbol{P}^T \boldsymbol{A} \boldsymbol{Q} \in \mathbb{R}^{l \times k}$, which is much smaller in size than $\boldsymbol{A}\boldsymbol{V} \in \mathbb{R}^{m \times k}$ in Algorithm 1 ($m > l$).

The procedure of the two-sided tensor-based algorithm is similar to the previously introduced tensor-based algorithm. First, reshape the training matrices to a third-order tensor $\mathcal{A}$. Then, obtain two sketching matrices $\boldsymbol{S}, \boldsymbol{W}$ by computing the Tucker2 decomposition of $\mathcal{A}$. Finally, taking $\boldsymbol{S}, \boldsymbol{W}$ and a test matrix $\boldsymbol{A}$ as input, use two-sided SCW to get the low-rank approximation of $\boldsymbol{A}$. We summarize this in Algorithm 4. Recall that we compute the Tucker2 decomposition of $\mathcal{A}$ by HOOI

(Lathauwer et al., 2000). If $k, l \sim \mathcal{O}(r)$, the time cost for Tucker2 decomposition with HOOI is $\mathcal{O}(rmnD' + r(m+n)D'^2)$, while Tucker1 decomposition costs $\mathcal{O}(mn^2D')$. In addition, as mentioned before, two-sided SCW is more efficient than the SCW algorithm. That means, the time complexity of the two-sided algorithm is asymptotic less than the original tensor-based algorithm because $r \ll m, n$, usually. However, since two-sided SCW uses $\boldsymbol{S}, \boldsymbol{W}$ to compress both the row and column space of $\boldsymbol{A}$ while the SCW compresses the row space only, there would be some loss in accuracy for the two-sided tensor-based algorithm compared to the tensor-based one.

---

**Algorithm 4** The two-sided tensor-based algorithm for low-rank approximation of the data stream $\mathcal{D}$.

---

**Input:** Test matrix $\boldsymbol{A}$, training set $\{\boldsymbol{A}_{d'} \in \mathbb{R}^{m \times n}\}_{d'=1}^{D'}$, target rank $r \leq \min\{m, n\}$, # rows of the sketching matrix $k$ and $l$.

1: Tensorization: $\boldsymbol{\mathcal{A}} \in \mathbb{R}^{m \times n \times D'} \leftarrow \{\boldsymbol{A}_{d'}\}_{d'=1}^{D'}$

2: $\boldsymbol{S}, \boldsymbol{W} \leftarrow$ Tucker2 decomposition of $\boldsymbol{\mathcal{A}}$ along the mode-1 and 2

3: $\hat{\boldsymbol{A}} \leftarrow$ Two-sided SCW($\boldsymbol{A}, \boldsymbol{S}, \boldsymbol{W}, r$)

**Output:** Low-rank approximation of $\boldsymbol{A}$: $\hat{\boldsymbol{A}}$

---

## 4 NUMERICAL EXPERIMENTS

In this section, we test our algorithms and compare them to the existing data-driven algorithms for low-rank approximation of data streams. We use three datasets for comparison — HSI (Imamoglu et al., 2018), Logo (Indyk et al., 2019) and MRI.

Table 1: Summary of datasets used for experiment.

| Name | Description | Dimension | #Train | #Test |
|------|-------------|-----------|--------|-------|
| HSI[1] | Hyper spectral images | $1024 \times 768$ | 100 | 400 |
| Logo[2] | Video | $3240 \times 1920$ | 100 | 400 |
| MRI[3] | Magnetic resonance imaging | $217 \times 181$ | 30 | 120 |

We measure the quality of the sketching matrix $\boldsymbol{S}$ by the error on the test set, and the test error is defined as Error $= \frac{1}{|\mathcal{D}_{\text{test}}|} \sum_{\boldsymbol{A} \in \mathcal{D}_{\text{test}}} \frac{\|\boldsymbol{A} - \hat{\boldsymbol{A}}\|_F - \|\boldsymbol{A} - \boldsymbol{A}_{\text{opt}}\|_F}{\|\boldsymbol{A} - \boldsymbol{A}_{\text{opt}}\|_F}$, where $\boldsymbol{A}_{\text{opt}}$ is the best rank-$r$ approximation of $\boldsymbol{A}$, and $\hat{\boldsymbol{A}}$ is the low-rank approximation computed by the tested algorithms. In all experiments, we set the rank $r$ to 10, and the sketching size $k = l = 20$. Experiments are run on a server equipped with an NVIDIA Tesla V100 card.

**Baselines.** As baselines, three methods are included — IVY (Indyk et al., 2019), Few-Shot (Indyk et al., 2021), and Butterfly (Ailon et al., 2021).

*IVY.* As described in Ref. (Indyk et al., 2019), the sketching matrix is initialized by a sign matrix. Its non-zero values are optimized by stochastic gradient descent (SGD) (Saad, 1998), which is an iterative optimization method widely used in machine learning.

*Few-Shot.* In (Indyk et al., 2021), as IVY does, the sketching matrix is sparse, and the location of the non-zero entries is fixed. The non-zero values of the sketch are also optimized by SGD. They proposed one-shot closed-form algorithms (including *1Shot1Vec+IVY* and *1Shot2Vec*), and the FewShotSGD algorithm with either 2 or 3 randomly chosen training matrices (i.e., *FewShotSGD-2* and *FewShotSGD-3*). We compare our algorithms with all of them.

*Butterfly.* In (Ailon et al., 2021), it is proposed to replace a dense linear layer in a neural network by the butterfly network. They suggested using a butterfly gadget for learning the low-rank

---

[1]Retrieved from `https://github.com/gistairc/HS-SOD`.
[2]Retrieved from `http://youtu.be/L5HQoFIaT4I`.
[3]Retrieved from `https://brainweb.bic.mni.mcgill.ca/cgi/brainweb2`.

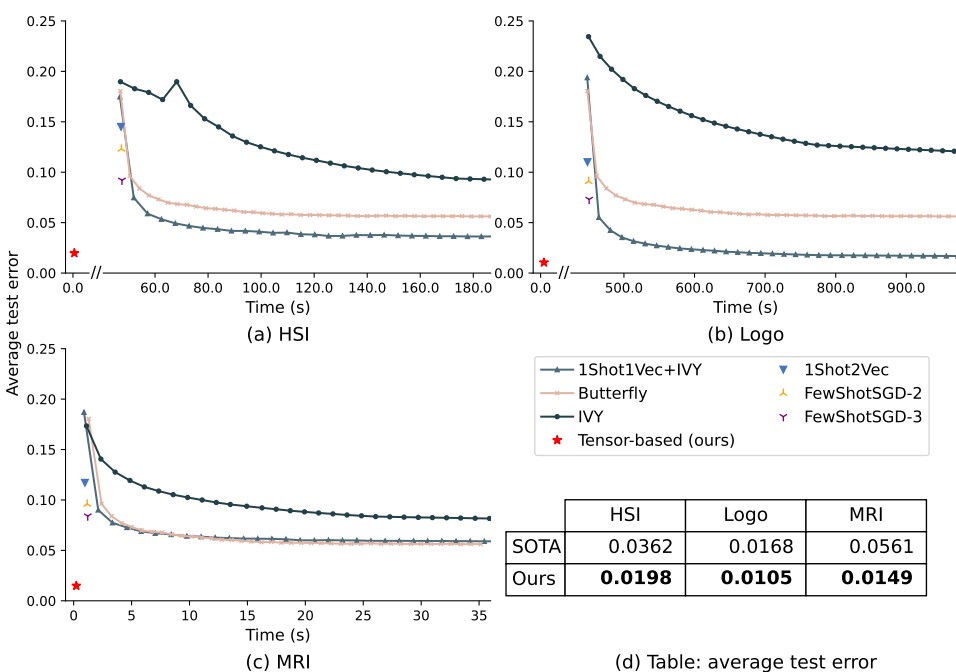

Figure 1: Test error per training time with the target rank $r = 10$ and the sketching size $k = 20$. In (d), SOTA represents the lowest test error that the baselines achieve.

approximation, also learning the non-zero values of a sparse sketching matrix by SGD, similarly to IVY.

Since the baselines above all use one sketching matrix only, we compare our tensor-based algorithm with them. For the two-sided tensor-based algorithm, we test its performance later in this section, only comparing it with our tensor-based algorithm.

**Training time and test error.** We compare the test error per training time for each approach. The results are reported in Figure 1. Table in (d) in Figure 1 lists the test error of the tensor-based algorithm and the lowest test error among the baselines. The tensor-based algorithm achieves at least **0.55/0.64/0.27** times lower test error on HSI/Logo/MRI than the baselines. For the baselines, the training matrices are required to be normalized to avoid the imbalance in the dataset before the training starts. On HSI/Logo/MRI, this pre-processing of data takes **46.55s/447.67s/0.72s**. After that, the training for the sketching matrix could start. However, for our algorithms, this pre-processing time can be avoided, because our algorithms are to compute the top-$r$ subspace of the training matrices which remains when the training data scales by a constant. On HSI/Logo/MRI, the tensor-based algorithm takes **0.53s/4.76s/0.23s** for training, which is much faster than *IVY, 1Shot1Vec+IVY* and *Butterfly*. As a result, our algorithm significantly outperforms the baselines — much more accurate and faster.

**Testing time.** Next we report running time for the testing process on all datasets. Note that the sketching matrix by the tensor-based algorithm is dense, while that of the baselines is sparse. This results in the difference in the testing process, mainly on the matrix multiplication $SA$ in the SCW procedure. The baselines have approximately the same testing time since their sketching matrix has the same sparsity. For the tensor-based algorithm, we use the built-in functionality in PyTorch, i.e., $\mathrm{torch.matmul}(S, A)$, to compute $SA$, which provides great acceleration. On HSI/Logo/MRI, the testing process for the tensor-based algorithm takes **0.52s/0.69s/0.28s**. For the baselines, the sketching matrix $S$ is sparse and $S$ is stored using two vectors — one for storing the location of non-zero entries, and the other for storing the values of the non-zero entries. Suppose that the location and value vectors are $l$, $v$. To compute $SA$, we can update $SA[l_i, j] = SA[l_i, j] + v_i A_{ij}$. In this way, the testing time for the baselines is **19.31s/60.15s/1.71s** on HSI/Logo/MRI, which is much longer than that of the tensor-based algorithm, mainly because there is no software acceleration

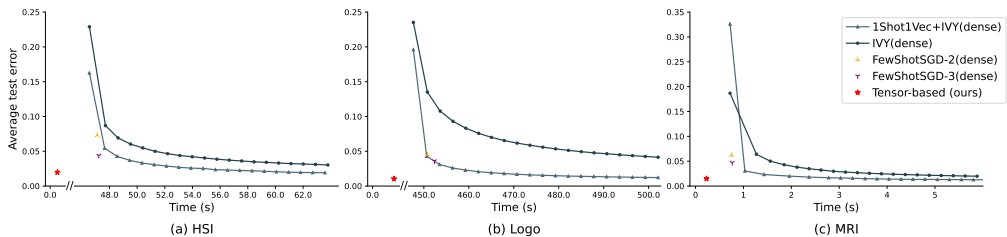

Figure 2: Test error per training time compared with the dense baselines. Note that the training of the dense baselines is faster than the original baselines for the use of $\mathrm{torch.matmul}()$ to compute $\boldsymbol{SA}$.

applied here. However, the testing process for the tensor-based algorithm is accelerated by the use of the built-in functionality $\mathrm{torch.matmul}()$ in PyTorch. If we restore the sparse $\boldsymbol{S}$ in the baselines as the COO format and use $\mathrm{torch.sparse.mm}(\boldsymbol{S}, \boldsymbol{A})$ to compute $\boldsymbol{SA}$ when testing, the testing time for the baselines on HSI/Logo/MRI is reduced to **0.88s/1.17s/0.40s**.

In view of the dense structure of our sketching matrix, we implement the baselines as dense ones and compare our algorithm with them, including *IVY (dense)*, *1Shot1Vec+IVY (dense)*, *FewShotSGD-2 (dense)* and *FewShotSGD-3 (dense)*. The dense baselines optimize all entries of the sketching matrix, not only the non-zero entries as the original baselines did. We show the results in Figure 2. The test error for the tensor-based algorithm is **0.0198/0.0105/0.0149** on HSI/Logo/MRI, while the lowest test error that the dense baselines achieve is **0.0191/0.0122/0.0124** on HSI/Logo/MRI. With comparable accuracy, our approach has much shorter training time than the dense baselines.

**Experiments for the two-sided algorithm.** Finally, we test the performance of the two-sided tensor-based algorithm. This algorithm uses two sketching matrices $\boldsymbol{S}, \boldsymbol{W}$ for computing the low-rank approximation. Table 2 shows the test error, the training time and the testing time for the two proposed algorithms, the tensor-based algorithm and two-sided version. The tensor-based algorithm achieves **0.29/0.73/0.63** times lower test error on HSI/Logo/MRI than the two-sided algorithm. However, the two-sided algorithm has both shorter training time and testing time. These results confirm our analysis in Section 3.

Table 2: Test error, training time and testing time of the tensor-based algorithm and the two-sided tensor-based algorithm.

| Datasets | Algorithms | test error | training time (s) | testing time (s) |
|---|---|---|---|---|
| HSI | tensor-based | 0.020 | 0.53 | 0.52 |
| | two-sided | 0.069 | 0.39 | 0.41 |
| Logo | tensor-based | 0.011 | 4.76 | 0.69 |
| | two-sided | 0.015 | 1.36 | 0.50 |
| MRI | tensor-based | 0.015 | 0.23 | 0.28 |
| | two-sided | 0.024 | 0.17 | 0.12 |

**Additional experiments.** In the experiments above, we only evaluate the algorithms when the sample ratio for training is 20% ($\frac{100}{100+400} = \frac{100}{100+400} = \frac{30}{30+120} = 20\%$ for HSI/Logo/MRI, respectively), which we denote as $\mathrm{sample\_ratio} = 20\%$. Figure 3 shows the performance of our proposed algorithms under different values of $\mathrm{sample\_ratio}$, including $2\%, 20\%$ and $80\%$. The results show that using only a small number of training matrices ($\mathrm{sample\_ratio} = 2\%$ for example), our algorithms achieve low enough error. When $\mathrm{sample\_ratio}$ increases from $2\%$ to $80\%$, the test error of the tensor-based algorithm decreases by a multiplicative factor of **0.952/0.917/0.684** on HSI/Logo/MRI, and for the two-sided tensor-based algorithm, the corresponding factor is **0.873/0.789/0.606** on HSI/Logo/MRI. On MRI, the test error decreases more when the number of training matrices increases compared to the other two datasets. In our opinion, this is because there is less stronger

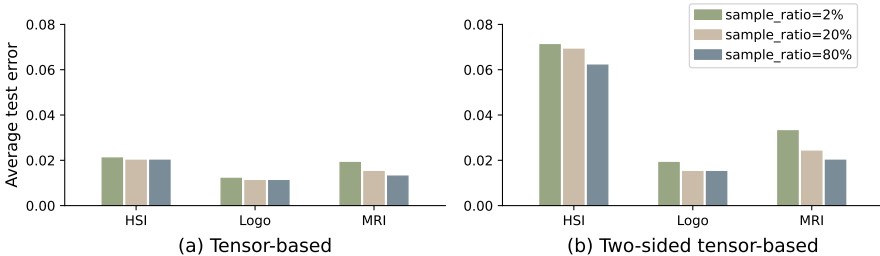

Figure 3: Test error of the tensor-based algorithm and the two-sided tensor-based algorithm under different number of training samples.

relevance among matrices of MRI. In general, increasing training samples improves the accuracy, but not significantly.

For our approach, one of the limitations is that it is required to load the whole training tensor at once. But for the baselines, the sketching matrix is learned by SGD and one of its advantages is that only a few (batch-size) training matrices are required to load in memory at a time. However, the results in Figure 3 show that a small number of training matrices are enough to achieve good low-rank approximation for both the tensor-based algorithm and the two-sided tensor-based algorithm. As a result, the memory usage of the proposed algorithms is also relatively low, comparable to the baselines.

## 5 CONCLUSIONS AND FUTURE WORK

In this work, we propose an efficient and accurate approach to deal with low-rank approximation of data streams, namely the tensor-based sketching method. From a subspace perspective, we develop a tensor-based algorithm as well as a two-sided tensor-based algorithm. Numerical experiments show that the two-sided tensor-based algorithm is faster but attains higher test error than the tensor-based algorithm. Compared to the baselines, both algorithms are not only more accurate, but also far more efficient.

This work mainly focuses on reducing the training time for generating the sketching matrix. However, reducing the testing time is also of great interest. One of the approaches is to develop pass-efficient sketching-based algorithms for low-rank approximation. In applications, the pass-efficiency becomes crucial when the data size exceeds memory available in RAM. Further, in addition to low-rank approximation, the idea of the tensor-based sketching method can be applied to more operations such as $\varepsilon$-approximation and linear system solutions on data streams. We leave them for future work.

### ACKNOWLEDGMENTS

The work was supported by the High-performance Computing Platform of Peking University. The authors acknowledge it for supporting the computational work sincerely.

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

# A  APPENDIX

## A.1  PROOF OF THEOREM 1

**Proof.** The inequality in 3.4 will be proved if we prove the following two inequalities.

$$\sum_{d'=1}^{D'} \|\boldsymbol{A}_{d'} - \text{SCW}(\boldsymbol{S}, \boldsymbol{A}_{d'})\|_F^2 \leq \sum_{d'=1}^{D'} \|\boldsymbol{A}_{d'} - \boldsymbol{S}^T[\boldsymbol{S}\boldsymbol{A}_{d'}]_r\|_F^2, \tag{A.1}$$

and

$$\sum_{d'=1}^{D'} \|\boldsymbol{A}_{d'} - \boldsymbol{S}^T[\boldsymbol{S}\boldsymbol{A}_{d'}]_r\|_F^2 \leq \|\boldsymbol{\mathcal{A}}\|_F^2 - \|[\boldsymbol{S}\boldsymbol{A}_{(1)}]_r\|_F^2. \tag{A.2}$$

First, we consider the inequality in A.1. Let $\boldsymbol{Q} \in \mathbb{R}^{n \times k}$ be a column-wise orthogonal matrix in the row space of $\boldsymbol{S}\boldsymbol{A}_{d'}$. By definition of SCW, we have

$$\|\boldsymbol{A}_{d'} - \text{SCW}(\boldsymbol{S}, \boldsymbol{A}_{d'})\|_F^2 = \|\boldsymbol{A}_{d'} - [\boldsymbol{A}_{d'}\boldsymbol{Q}]_r\boldsymbol{Q}^T\|_F^2 = \|\boldsymbol{A}_{d'}\|_F^2 - \|[\boldsymbol{A}_{d'}\boldsymbol{Q}]_r\|_F^2. \tag{A.3}$$

Similarly,

$$\|\boldsymbol{A}_{d'} - \boldsymbol{S}^T[\boldsymbol{S}\boldsymbol{A}_{d'}]_r\|_F^2 = \|\boldsymbol{A}_{d'}\|_F^2 - \|[\boldsymbol{S}\boldsymbol{A}_{d'}]_r\|_F^2. \tag{A.4}$$

Combing A.3 and A.4, A.1 follows immediately if we show

$$\|[\boldsymbol{S}\boldsymbol{A}_{d'}]_r\|_F \leq \|[\boldsymbol{A}_{d'}\boldsymbol{Q}]_r\|_F. \tag{A.5}$$

Noting that

$$\boldsymbol{S}\boldsymbol{A}_{d'}\boldsymbol{Q} = \boldsymbol{U}\boldsymbol{\Sigma}\boldsymbol{V}^T\boldsymbol{Q} = \boldsymbol{U}\boldsymbol{\Sigma}\boldsymbol{Q}',$$

where $\boldsymbol{U}\boldsymbol{\Sigma}\boldsymbol{V}^T$ is the singular value decomposition of $\boldsymbol{S}\boldsymbol{A}_{d'}$ and $\boldsymbol{Q}' = \boldsymbol{V}^T\boldsymbol{Q}$. Since $\boldsymbol{V}$ and $\boldsymbol{Q}$ lie in the same row space and are both column-wise orthogonal, it is easy to see that $\boldsymbol{Q}'$ is a $k$-dimensional orthogonal matrix. Thus, $\boldsymbol{S}\boldsymbol{A}_{d'}$ and $\boldsymbol{S}\boldsymbol{A}_{d'}\boldsymbol{Q}$ share the same singular values. Combining Cauchy interlace theorem (Carlson, 1983), we have

$$\|[\boldsymbol{S}\boldsymbol{A}_{d'}]_r\|_F = \|[\boldsymbol{S}\boldsymbol{A}_{d'}\boldsymbol{Q}]_r\|_F \leq \|[\boldsymbol{A}_{d'}\boldsymbol{Q}]_r\|_F, \tag{A.6}$$

which proves A.5.

We now turn to the inequality in A.2. For convenience, we rewritten $\boldsymbol{SA}_{(1)}$ and $[\boldsymbol{SA}_{(1)}]_r$ with block components as

$$\boldsymbol{SA}_{(1)} = [\boldsymbol{SA}_1|\boldsymbol{SA}_2|\cdots|\boldsymbol{SA}_{D'}],$$

and

$$[\boldsymbol{SA}_{(1)}]_r = [\boldsymbol{B}_1|\boldsymbol{B}_2|\cdots|\boldsymbol{B}_{D'}],$$

where $\boldsymbol{B}_i \in \mathbb{R}^{k \times n}$ for $i = 1, \cdots, D'$. We then have

$$
\begin{aligned}
\sum_{d'=1}^{D'} \|\boldsymbol{SA}_{d'}\|_F^2 - \sum_{d'=1}^{D'} \|[\boldsymbol{SA}_{d'}]_r\|_F^2 &= \sum_{d'=1}^{D'} \|\boldsymbol{SA}_{d'} - [\boldsymbol{SA}_{d'}]_r\|_F^2 \\
&\leq \sum_{d'=1}^{D'} \|\boldsymbol{SA}_{d'} - \boldsymbol{B}_{d'}\|_F^2 \\
&= \|\boldsymbol{SA}_{(1)}\|_F^2 - \|[\boldsymbol{SA}_{(1)}]_r\|_F^2,
\end{aligned}
$$

where the Eckart-Young theorem is applied. It follows that

$$\sum_{d'=1}^{D'} \|[\boldsymbol{SA}_{d'}]_r\|_F^2 \geq \|[\boldsymbol{SA}_{(1)}]_r\|_F^2,$$

which is equivalent to in A.2.

Hence, we have proved that $\|\boldsymbol{\mathcal{A}}\|_F^2 - \|[\boldsymbol{SA}_{(1)}]_r\|_F^2$ is a relaxation of $\sum_{\boldsymbol{A}_{d'} \in \mathcal{D}_{\mathrm{train}}} \|\boldsymbol{A}_{d'} - \mathrm{SCW}(\boldsymbol{S}, \boldsymbol{A}_{d'})\|_F^2$. Therefore, the problem 2.1 can be converted to minimize $\|\boldsymbol{\mathcal{A}}\|_F^2 - \|[\boldsymbol{SA}_{(1)}]_r\|_F^2$, i.e., maximize $\|[\boldsymbol{SA}_{(1)}]_r\|_F^2$. Due to $k > r$, it is not difficult to verify that a sufficient condition for maximizing $\|[\boldsymbol{SA}_{(1)}]_r\|_F^2$ is maximizing $\|\boldsymbol{SA}_{(1)}\|_F^2$, which is equivalent to 3.3. As a result, instead of optimizing problem 2.1, we can covert it to our proposed problem 3.3.

Hence, our proof is completed. $\qquad\qquad\square$

## A.2 Proof of Theorem 2

**Proof.** Let $\hat{\boldsymbol{U}}\hat{\boldsymbol{\Sigma}}\hat{\boldsymbol{V}}^T$ be the SVD of the matrix $\boldsymbol{SA}$. Using the definition of the SCW algorithm, we have $\mathrm{SCW}(\boldsymbol{A}, \boldsymbol{S}, r) = [\boldsymbol{A}\hat{\boldsymbol{V}}]_r \hat{\boldsymbol{V}}^T$. Further, since $\hat{\boldsymbol{V}}$ is column-wise orthogonal, we have

$$\|\boldsymbol{A} - \mathrm{SCW}(\boldsymbol{A}, \boldsymbol{S}, r)\|_F^2 = \|\boldsymbol{A} - [\boldsymbol{A}\hat{\boldsymbol{V}}]_r \hat{\boldsymbol{V}}^T\|_F^2 = \|\boldsymbol{A}\|_F^2 - \|[\boldsymbol{A}\hat{\boldsymbol{V}}]_r\|_F^2.$$

Similarly, we have

$$\|\boldsymbol{A} - [\boldsymbol{A}]_r\|_F^2 = \|\boldsymbol{A}\|_F^2 - \|[\boldsymbol{U}_k^T \boldsymbol{A}]_r\|_F^2.$$

Recall that $\boldsymbol{U}\boldsymbol{\Sigma}\boldsymbol{V}^T$ is the SVD of $\boldsymbol{A}$, and $\boldsymbol{U}_k$ be a matrix composed of the first $k$ columns of $\boldsymbol{U}$. Based on the result A.6 in the proof of Theorem 1, we immediately get

$$\|[\boldsymbol{A}\hat{\boldsymbol{V}}]_r\|_F^2 \geq \|[\boldsymbol{SA}]_r\|_F^2. \tag{A.7}$$

Thus, we have

$$
\begin{aligned}
\|\boldsymbol{A} - \mathrm{SCW}(\boldsymbol{A}, \boldsymbol{S}, r)\|_F^2 - \|\boldsymbol{A} - [\boldsymbol{A}]_r\|_F^2 &\leq \|[\boldsymbol{U}_k^T \boldsymbol{A}]_r\|_F^2 - \|[\boldsymbol{SA}]_r\|_F^2 \\
&\leq \|\boldsymbol{U}_k^T \boldsymbol{A}\|_F^2 - \|\boldsymbol{SA}\|_F^2 \\
&= \mathrm{tr}(\boldsymbol{A}^T (\boldsymbol{U}_k \boldsymbol{U}_k^T - \boldsymbol{S}^T \boldsymbol{S}) \boldsymbol{A}) \\
&\leq \|\boldsymbol{U}_k \boldsymbol{U}_k^T - \boldsymbol{S}^T \boldsymbol{S}\|_2^2 \|\boldsymbol{A}\|_F^2 \\
&\leq \|\boldsymbol{U}_k \boldsymbol{U}_k^T - \boldsymbol{S}^T \boldsymbol{S}\|_F^2 \|\boldsymbol{A}\|_F^2 \\
&\leq \mathcal{O}(\varepsilon) \|\boldsymbol{A}\|_F^2.
\end{aligned}
\tag{A.8}
$$

Hence, our proof is completed. $\qquad\qquad\square$

## A.3 HOOI ALGORITHM

---

**Algorithm 5** HOOI algorithm Lathauwer et al. (2000); Kolda & Bader (2009)

---

**Input:**

    Tensor $\boldsymbol{\mathcal{A}} \in \mathbb{R}^{I_1 \times I_2 \cdots \times I_N}$

    Truncation $(R_1, R_2, \cdots, R_N)$

    Initial guess $\{\boldsymbol{U}_0^{(n)} : n = 1, 2, \cdots, N\}$

**Output:**

    Low multilinear-rank approximation $\hat{\boldsymbol{\mathcal{A}}} = \boldsymbol{\mathcal{G}} \times_1 \boldsymbol{U}^{(1)} \times_2 \boldsymbol{U}^{(2)} \cdots \times_N \boldsymbol{U}^{(N)}$

1:   $k \leftarrow 0$

2: **while** not convergent **do**

3:     **for all** $n \in \{1, 2, \cdots, N\}$ **do**

4:         $\boldsymbol{\mathcal{B}} \leftarrow \boldsymbol{\mathcal{A}} \times_1 (\boldsymbol{U}_{k+1}^{(1)})^T \cdots \times_{n-1} (\boldsymbol{U}_{k+1}^{(n-1)})^T \times_{n+1} (\boldsymbol{U}_k^{(n+1)})^T \cdots \times_N (\boldsymbol{U}_k^{(N)})^T$

5:         $\boldsymbol{B}_{(n)} \leftarrow \boldsymbol{\mathcal{B}}$ in matrix format

6:         $\boldsymbol{U}, \boldsymbol{\Sigma}, \boldsymbol{V}^T \leftarrow$ truncated rank-$R_n$ SVD of $\boldsymbol{B}_{(n)}$

7:         $\boldsymbol{U}_{k+1}^{(n)} \leftarrow \boldsymbol{U}$

8:         $\boldsymbol{\mathcal{G}}_{k,n} \leftarrow \boldsymbol{\Sigma} \boldsymbol{V}^T$ in tensor format

9:         $k \leftarrow k + 1$

10:    **end for**

11: **end while**

12: $\boldsymbol{\mathcal{G}} \leftarrow \boldsymbol{\Sigma} \boldsymbol{V}^T$ in tensor format

---

