# OpenReview forum: "Tensor-Based Sketching Method for the Low-Rank Approximation of Data Streams."
_ICLR.cc/2023/Conference — ICLR 2023 poster_

### Official Review · Reviewer_337b · 2022-10-20

**Confidence:** 4
**Clarity, Quality, Novelty And Reproducibility:** See the last question for detail.
**Correctness:** 2
**Technical Novelty And Significance:** 2
**Empirical Novelty And Significance:** 3
**Recommendation:** 5

**Strength And Weaknesses:**

Strength

1. The paper proposes an algorithm from a new insight, which has not been considered in the literature on the learning-based LRA. The paper also gives a theoretical guarantee.

2. The experimental results are strong and interesting. The proposed algorithm has a lower error while having a faster training time.

Weakness

0. Note that the template of this paper is actually a little different than the ICLR template.
1. The analysis of the theorem is straightforward. Besides, the Theorem 2 seems only to give an additive error $\epsilon ||A||_F^2$,  I am wondering that would it be possible to get a multiplicative error $\epsilon ||A- A_k||_F^2$, which is the common setting in the literature of the sketching-based LRA.
2. As mentioned in the paper, the new learning-based sketch matrix is a dense matrix, which I think is the main drawback of this paper. The dense sketch matrix seems not to be useful when the size of the data is large. Note that if $S$ is a Count-Sketch matrix, we can compute $SA$ in $nnz(A)$ time. That is one of the reasons that most of the previous works only study the sparse sketching matrix.  As mentioned in the work of IVY19, One applications scenario of the learning-based sketch involves processing streams of data (video, data logs, customer activity etc) by executing the same algorithm on an hourly, daily or weekly basis. Hence in this sense the testing time is more important if the training time is only a little slower as the training process can be done before the new data comes. Also, notice that the sparse matrix has the pass-efficiency property when the data size exceeds memory available in RAM, which is what the author mentioned in the conclusion section. The update time for one update of the Count-Sketch matrix is $O(1)$.

3. The paper mentions the testing time in the experiment section but it seems not to be done in the correct way. It says in the paper that "have no idea how to accelerate the matrix multiplication SA while exploiting the sparse structure of S". However, suppose that the position and value vectors are $p$, $v$. Note that if the $t$-th non-zero entries of $A$ is $A_{ij} = c$, we just need to do the update
$
SA[p_i, j] = SA[p_i, j]  + v_i \cdot c .
$
which means the time to compute $SA$ is the same as to read the matrix $A$.

4. The authors mention the memory usage in the experiment section. Can the author give a detailed result of the memory usage for all approaches?

**Summary Of The Paper:**

The paper studies the learning-based low-rank approximation problem and proposes a new algorithm, which is based on the tucker tensor decomposition. The new optimization problem based on tensor decomposition can be seen as a relaxation of the original LRA problem. The experimental result suggests that the proposed algorithm has a lower error while having a faster training time.

**Summary Of The Review:**

Overall I think the approach the author proposed is interesting. My main concern is the non-sparse of the sketch matrix so currently I vote as a 5  the paper.  I can raise the score if the author can address this adequately (note that the sparsity property is not the only way to compute $SA$ fast, like the Subsampled Randomized Hadamard Transform(SRHT) matrix is a dense sketching matrix while we can compute $SA$ in $O(nd \log n)$ time) .

---

### Official Review · Reviewer_5H7S · 2022-10-24

**Confidence:** 4
**Correctness:** 3
**Technical Novelty And Significance:** 2
**Empirical Novelty And Significance:** 2
**Recommendation:** 6

**Clarity, Quality, Novelty And Reproducibility:**

Clarity: I found the proofs abbreviated and difficult to follow, and it would help to expand them and add explanations and bridging steps.
Reproducibility: I appreciate the fully disclosed details of the implementation.

**Strength And Weaknesses:**

The tensor-based approach seems interesting and may lead to useful insights on the problem. However, there is a major issue that makes the proposed algorithm incomparable to prior work, both theoretically and empirically - the lack of sparsity of the output sketching matrix. This unfortunately renders the paper methodologically unviable.

In all prior work (both oblivious and data-driven), the sketching matrix has a certain structure that allows fast matrix multiplication by it (usually sparsity), while here no such structure is guaranteed. On the theoretical level, this makes the resulting low-rank approximation algorithm asymptotically slower than all of the prior work it is compared to.

The manuscript is aware of this issue, and finally addresses it on the empirical level, on page 8, stating that even though the asymptotic running time is slower, the proposed algorithm is in fact empirically faster than prior work based on experiments. It goes on to explain this is because the implementation of the proposed algorithm uses the built-in (highly optimized) PyTorch matrix multiplication functionality, while the baselines were implemented with the non-optimized (and inevitably much slower) sparse matrix multiplication code released by the authors of those works. This unfortunately renders the reported testing time results essentially meaningless: the algorithms should be compared in the same experimental setting (software and hardware). Using the same dense matrix multiplication functionality in PyTorch for the sparse baselines would have already improved their running time to the same as the proposed method, erasing any reported advantage in testing time.

To be clear, it is true that as of now, PyTorch and comparable optimized software packages for matrices do not exploit sparsity, and dense matrix multiplication is as fast as sparse on the datasets considered here. Sparsity is useful theoretically (for getting better asymptotic bounds) and also in practice if the matrices are too large to fit in working memory (which makes dense matrix multiplication slow or infeasible). I do not question the reported results themselves, and one could perhaps make the argument that the sparsity of the sketching matrix is not helpful at all and could be dispensed with, but the current framing and methodological design of the paper are unviable. If sparsity is indeed unhelpful, then the sparse baseline should be implemented as dense ones and not crippled by attempting to leverage sparsity, and dense methods for low-rank approximation could be considered and reported as well (even full SVD).

**Summary Of The Paper:**

The paper studies the data-driven low-rank approximation problem on data streams, which has gained attention recently. For a single matrix, some fast algorithms for low-rank approximation are based on sketching, and specifically they choose a sparse sketching matrix with random signs, which is multiplied by the original matrix to reduce its row and/or column dimensions while still allowing approximate recovery of the best low-rank approximation. In a stream of matrices assumed to be related to each other, it has been recently shown that the entries of the sketching matrix can be chosen in a data-driven way based on matrices previously seen on the stream instead of using random signs, in order to improve performance.

This paper suggests a new algorithm along these lines, that views the stream of matrices as a tensor of order 3, which leads to a formulation of a loss problem that can be solved directly and yields an appropriate sketching matrix, albeit non-sparse.



**Summary Of The Review:**

This is a potentially interesting paper but with flaws in its framing and methodological design, with which I cannot recommend acceptance. I could have recommended accepting a version of the paper with largely the same technical material, but that clarifies properly the departure from prior work, and with a viable experimental design that compares the baselines on a level ground and includes more relevant baselines. I also recommend rewriting the proofs in a clearer way that would assist in verifying the stated theorems.

---

### Official Review · Reviewer_dbkp · 2022-10-27

**Confidence:** 3
**Correctness:** 3
**Technical Novelty And Significance:** 4
**Empirical Novelty And Significance:** Not applicable
**Recommendation:** 6

**Clarity, Quality, Novelty And Reproducibility:**

The quality is fine. Some definitions are not so clear, e.g. in (1.1) where the authors define B, is this B for all d, or it should be B_d for different d? It is pretty novel.

**Strength And Weaknesses:**

Strength: in those previous algorithms, they require the approximation for matrices in the training set to be as accurate as possible and this work provides the method based on Tucker decomposition with theoretical analysis. And the idea of viewing a stream of matrices as a tensor is novel and helpful in finding their common subspaces.

Weakness: the analysis is not complete. It didn’t mention how close is S, the estimator of U_k, in general cases to U_k. Also the bound in theorem 2 is not clear with m and n.


**Summary Of The Paper:**

In this work, the authors summarized previous algorithms to recover column/row spaces of a set of matrices fast and introduced Tucker decomposition of tensor into them, which can give a good choice of hyperparameter in those algorithms.

**Summary Of The Review:**

This work should qualify for a conference paper, although there are something which can be improved.

---

### Official Review · Reviewer_8sdZ · 2022-10-31

**Confidence:** 3
**Correctness:** 4
**Technical Novelty And Significance:** 3
**Empirical Novelty And Significance:** 2
**Recommendation:** 6

**Clarity, Quality, Novelty And Reproducibility:**

Clarity: paper is well written.

Quality: the paper has the right 'ingredients' and is of good quality overall.

Novelty: the algorithm proposed is an extension of existing methods -- see above.

Reproducibility: the authors do not provide code or mention the intention of sharing their code in case of acceptance.

**Strength And Weaknesses:**

+

* The paper is well written and clear for the most part. It does need to be proof read for small typos and grammar.
*  The idea is well grounded in existing methods, e.g. IVY and SCW and the approximation quality is derived in two theorems.
*  The authors test their algorithm on real datasets and show some improvements.

-
* Overall I have some concerns regarding the novelty as the proposed algorithms are somewhat straightforward extensions of the existing algorithms, e.g. SCW.

**Summary Of The Paper:**

The paper introduces a low-rank tensor approximation algorithm intended for data streams applications.
According to the authors, the algorithm fully exploits the structure of data streams and obtains quasi-optimal sketching matrices by performing tensor decomposition on training data.
The approach computes the sketching matrix by minimizing a new loss function which is a relaxation of that in the IVY tensor decomposition method. Specifically, the key is that the minimization of this loss function by tensor decomposition on the training set is non-iterative. The authors formulate a couple of theorems that bound the approximation performance and provide empirical analysis on several real data stream datasets.

**Summary Of The Review:**

In summary, I am weakly included to accept this paper given that it scores high in terms of clarity and quality. However, my inclination is weak due to some minor concerns regarding its novelty.

---

### Decision · Program_Chairs · 2023-01-20

**Decision:**

Accept: poster

**Justification For Why Not Higher Score:**

This is more or less a borderline paper -- the reviewers show less enthusiasm to recommend it as an acceptance with oral or spotlight.

**Justification For Why Not Lower Score:**

The authors have addressed most of the concerns of the referees.

**Metareview: Summary, Strengths And Weaknesses:**

This paper proposes a new tensor-based sketching method to achieve the low-rank approximation of data streams. The methods are sound and the contributions are novel. The referees raised a series of concerns and the authors have addressed most of them. A remaining concern is in the Section "Testing time" on Pages 7-8, which the authors shall address more carefully in their camera-ready version.

**Note From Pc:**

if the above contains the word "oral" or "spotlight" please see: "oral" presentation means -> notable-top-5% and "spotlight" means -> notable-top-25%. As stated in our emails, we are disassociating presentation type from AC recommendations

**Summary Of Ac-Reviewer Meeting:**

I, as the AC, had a virtual meeting with 4 referees. The major concern of Reviewers 5H7S and 337b were the claims of this paper about time cost and its comparison with existing methods was slightly unfair. After the rebuttal, Reviewer 5H7S responded to the author's rebuttal and raised their score. I still urge the authors to address this concern more carefully before publishing.